# Biomarkers Linked with Dynamic Changes of Renal Function in Asymptomatic and Mildly Symptomatic COVID-19 Patients

**DOI:** 10.3390/jpm11050432

**Published:** 2021-05-19

**Authors:** Ya-Chieh Chang, Ping-Huang Tsai, Yu-Ching Chou, Kuo-Cheng Lu, Feng-Yee Chang, Chia-Chao Wu

**Affiliations:** 1Department of Internal Medicine, Division of Nephrology, Taoyuan Armed Forces General Hospital, Taoyuan 325, Taiwan; ajie1124@gmail.com; 2Department of Internal Medicine, Division of Nephrology, Tri-Service General Hospital, National Defense Medical Center, Taipei 114, Taiwan; tsaipinghuang@gmail.com; 3School of Public Health, National Defense Medical Center, Taipei 114, Taiwan; trishow@mail.ndmctsgh.edu.tw; 4Department of Medicine, Division of Nephrology, Taipei Tzu Chi Hospital, Buddhist Tzu Chi Medical Foundation, New Taipei City 231, Taiwan; kuochenglu@gmail.com; 5Department of Internal Medicine, Division of Infectious Diseases and Tropical Medicine, Tri-Service General Hospital, National Defense Medical Center, Taipei 114, Taiwan; fychang@ndmctsgh.edu.tw; 6Department and Graduate Institute of Microbiology and Immunology, National Defense Medical Center, Taipei 114, Taiwan

**Keywords:** COVID-19, asymptomatic, C-reactive protein, neutrophil-to-lymphocyte ratio, renal function

## Abstract

The catastrophic coronavirus disease 2019 (COVID-19) pandemic is currently a critical global issue. One well-known complication of COVID-19 in severe cases is acute kidney injury, but no research has given a description of its impact on the kidney in patients with mild symptoms. We explore the renal function changes in mild COVID-19 patients. This retrospective, single-center study included 27 participants with laboratory-detected severe acute respiratory syndrome coronavirus two (SARS-CoV-2) infection who were admitted to the Tri-Service General Hospital from 4 February to 26 May 2020 and analyzed their clinical features, radiological findings, and laboratory data. Data collected upon admission and discharge showed a median estimated glomerular filtration rate (eGFR) of 106.7 mL/min/1.732 m^2^ and 112.2 mL/min/1.732 m^2^, respectively, with a *p*-value of 0.044. A correlation between renal function and the severity of infection was also found and was statistically significant upon admission. Patients with a lower lymphocyte count or higher C-reactive protein, neutrophil count, and neutrophil-to-lymphocyte ratio presented with a decreased eGFR during their early infection phase. The biomarkers (CRP and NLR) may be linked with dynamic changes of renal function in COVID-19 patients who are asymptomatic or have mild symptoms.

## 1. Introduction

The outbreak of coronavirus disease 2019 (COVID-19), caused by severe acute respiratory syndrome coronavirus two (SARS-CoV-2), has spread throughout most countries worldwide. Since late 2019, the COVID-19 pandemic has become a rising global health emergency and the greatest challenge we have encountered since World War II. It is one of our most vulnerable moments in history. Up to 1 February 2021, more than 100 million people have been infected and nearly 2.3 million killed globally by this virus. Studies have been aimed at exploring its epidemiology, pathophysiology, and disease characteristics. The mortality rate of COVID-19 is approximately 2.2% worldwide, and much evidence indicates prominent adverse effects from COVID-19 in Asian people [1,2]. In some countries, many people cannot afford COVID-19 medical care and have to choose home isolation unless in critical condition. Taiwan has been successful in handling the coronavirus pandemic. As of 1 February 2021, only 912 confirmed cases and 8 deaths have been reported in Taiwan. All patients with confirmed SARS-CoV-2 infection must receive compulsory hospital treatment even if they have mild illness.

SARS-CoV-2 binds to the angiotensin-converting enzyme 2 (ACE2) receptor for cellular entry, which is abundant on the surface of type II alveolar cells of the lung [3]. Thus, it is reasonable to say that COVID-19 primarily manifests as a pulmonary disorder with symptoms ranging from mild upper respiratory infection to severe pneumonia, pulmonary fibrosis, and acute respiratory distress syndrome. The kidney also highly expresses ACE2, which may explain the growing body of evidence displaying frequent kidney involvement among COVID-19 patients [4]. Acute kidney injury (AKI) has been delineated as a severe complication of COVID-19 with a higher risk of mortality. A large observational study by Hirsch JS et al. reported a 36.6% prevalence of AKI [5]. Prerenal azotemia and acute tubular necrosis are among the clinical characteristics. High incidences of proteinuria and hematuria have also been noted by dipstick testing [6]. Several suspected risk factors for AKI include severity of COVID-19, ventilator support, male sex, older age, comorbidity, and ethnicity [5,7].

COVID-19 patients have lower disease severity in Taiwan. According to Taiwan Centers for Disease Control (CDC) data, about 60% to 70% of COVID-19 patients are asymptomatic or have mild symptoms [8]. AKI mostly occurs in critical cases, but little is known whether COVID-19 affected renal function in patients with mild symptoms. Therefore, we performed this retrospective study to explore renal function in mild COVID-19 patients.

## 2. Materials and Methods

### 2.1. Study Design and Patients

We conducted this retrospective, single-center study at the Tri-Service General Hospital (TSGH), a medical center in northern Taiwan. Individuals with suspected travel history or contact history with clinical manifestations had to undergo further screening. Laboratory testing for SARS-CoV-2 infection was performed using real-time reverse-transcriptase polymerase chain reaction (RT-PCR) assays of nasopharyngeal swab specimens. Most patients received high-resolution computed tomography (HRCT) to confirm pulmonary involvement. This study assessed COVID-19 patients admitted to this hospital from 4 February to 26 May 2020. We aimed to investigate the effect of SARS-CoV-2 infection on the renal function of asymptomatic patients or those with only mild symptoms. We used the CURB-65 score to classify the severity of pneumonia, which includes 5 risk factors: confusion at new onset, >19 mg/dL blood urea nitrogen (BUN), >30 breaths per minute respiratory rate, <90 mmHg systolic and <60 mmHg diastolic blood pressure, and age >65. Each risk factor was assigned 1 point, for a maximum score of 5. Because those with scores ranging from 3 to 5 were classified as having severe pneumonia, we only enrolled patients with a score of ≤2. Thus, 27 patients, all of whom were Taiwanese, were enrolled, whereas 1 patient was excluded because of progressive pulmonary fibrosis and respiratory failure with ventilator support. According to Taiwan CDC guidelines, the discharge criteria include clinical improvement with radiographic resolution and 3 consecutive negative virus RT-PCR results.

### 2.2. Clinical Evaluation

The epidemiological and clinical characteristics, together with laboratory data, were extracted from the cohort. We reviewed the electronic medical records, laboratory results, and radiographic findings of all admitted patients with mild but laboratory-confirmed COVID-19. Laboratory examinations encompassed complete blood count, differential count, liver function test, BUN, creatinine, sodium, potassium, albumin, and C-reactive protein (CRP). We compared the variability of these data collected at 2 separate time points: within 24 h after admission and within 72 h prior to discharge.

We assessed the estimated glomerular filtration rate (eGFR) using the Modification of Diet in Renal Disease (MDRD) study equation, which is the most commonly used method worldwide: 186 × (creatinine/88.4) − 1.154 × (age) − 0.203 × (0.742 if female) × (1.210 if black). The AKI criteria are in accordance with the definition in the 2012 Kidney Disease: Improving Global Outcomes (KDIGO) Clinical Practice Guideline. Briefly, AKI is defined as a ≥0.3 mg/dL increase in serum creatinine over 48 h equivalent to a ≥50% relative increase in serum creatinine over 7 days.

### 2.3. Statistical Analysis

Statistical analysis was accomplished using SPSS software for Windows, version 20.0 (IBM Corp, Armonk, NY, USA). Descriptive analyses were performed for demographic, clinical, and laboratory data. We presented the results as median (interquartile range, IQR) for continuous measurements because of non-normal distribution, and categorical variables were reported as numbers and percentages (%). *p* < 0.05 was considered statistically significant.

## 3. Results

### 3.1. Study Population and Patient Characteristics

Detailed demographic information, comorbidities, and symptoms of all patients were recorded upon admission. A total of 27 laboratory-confirmed COVID-19 cases were included in this study, and all patients were discharged. The median age was 34 years, and 15 (55.6%) patients were male (Table 1). The median body mass index (BMI) was 21.8 kg/m^2^. There were no significant changes in systemic hemodynamics throughout our study. Of the 27 patients, 29.6% had one or more comorbidities. Among them, the most common comorbidities were dyslipidemia (25.9%), hypertension (7.4%), and diabetes (7.4%). Upon admission, most patients had cough (85.2%), fever (62.9%), and rhinorrhea (44.4%). Only one-third of the patients had shortness of breath (33.3%), and five were asymptomatic. Otherwise, some patients presented with atypical symptoms, including abnormal sense of smell (40.7%), diarrhea (37.0%), and abnormal sense of taste (29.6%). Other symptoms included sore throat, chillness, myalgia, chest pain, headache, nausea, and vomiting.

### 3.2. Disease Severity and Complications

We evaluated the disease severity by CURB-65. A total of 23 (85.2%) patients had a CURB-65 score of 0; 3 (11.1%), a score of 1; and 1 (3.7%), a score of 2 (Table 2). They were classified as mild severity according to the CURB-65 guidelines. Of the 27 patients, 88.9% underwent HRCT. Ground-glass opacity was noted in 44.4% of patients from HRCT, with the other 55.5% without obvious pneumonia. Asymptomatic patients only received isolation care, without any medication. Most of the other 22 patients were treated with macrolides and quinine. Some patients received transient third-line antibiotics due to marked ground-glass opacification of the lung on chest radiography. The median hospital stay was 28 days, with the majority of patients not developing complications, the most common of which was hepatitis, occurring in 14.8% of cases. No enrolled patient presented with multiple organ dysfunction, disseminated intravascular coagulation, pulmonary fibrosis, or AKI. The asymptomatic patients did not undergo any treatment but only received isolation care and monitoring (18.5%). Most of the other 22 patients were treated with chloroquine (74.0%) and macrolides (55.5%). Some patients received transient third-line antibiotics due to marked ground-glass opacification of the lung on chest radiography (Table 3).

### 3.3. Comparison of Laboratory Findings upon Admission and Discharge

Upon admission, all patients had a complete blood count and differential blood count test. The median results were as follows: white blood cell count, 4760 cells/µL; hemoglobin, 14.3 g/dL; platelet count, 230,000 cells/µL; neutrophil count, 65.8%; and lymphocyte count, 25.3%. Furthermore, the median neutrophil count and lymphocyte count were 57.3% and 31.7%, respectively, at discharge (Table 4). In this section, the decrease of neutrophil count and increase of lymphocyte count were statistically significant, which may indicate improvement of infection control during hospitalization. As for the biochemical profile, the median CRP was 0.39 mg/dL, and the median liver function test was elevated during hospitalization. The median initial aspartate transaminase and alanine aminotransferase levels were 18 U/L and 15 U/L, respectively, and then increased to 17 U/L and 20 U/L, respectively, within 3 days prior to discharge.

### 3.4. Changes in Renal Function and Correlation with the Severity of Infection

Data collected upon admission and discharge revealed an obvious improvement of renal function during the patients’ hospital stays. The median eGFRs were 106.7 mL/min/1.732 m^2^ and 112.2 mL/min/1.732 m^2^, respectively (Figure 1). The correlation between renal function and the severity of infection was statistically significant upon admission (Figure 2). We were able to observe a decline in eGFR when the neutrophil count, neutrophil-to-lymphocyte ratio (NLR), and CRP were elevated. Conversely, an increase in eGFR and lymphocyte count was noticed in our study. However, no significant correlation was noted between renal function and infection markers at discharge. We also integrated the role of CRP and NLR as predictive markers in COVID-19. A majority of prior studies have demonstrated positive correlations with disease severity or mortality (Table 5 and Table 6).

Seven patients had downregulation of eGFR on their day of discharge. Of these patients, four had a CURB-65 score of 0; two, a score of 1; and one, a score of 2. One patient, an 81-year old man, had diabetic nephropathy and chronic kidney disease, stage 3 primarily. Progressive reduction of renal function during this infection episode was observed. The other six patients were found to have slightly elevated creatinine levels (0.6 to 0.7 mg/dL in four and 0.7 to 0.8 mg/dL in two).

## 4. Discussion

Our study demonstrated a slight male predominance, with most patients being young adults or of middle age. The median BMI was within normal range, with less than one-third of the cases having one or more comorbidities. Based on the CURB-65 classification, all enrolled patients had mild disease severity. After approximately 1 month of treatment, these 27 patients were discharged with few complications. A significant improving trend was observed in renal function and infection parameters during hospitalization.

Despite the respiratory and immune systems being the crucial targets of SARS-CoV-2, prominent acute proximal tubular injury, peritubular erythrocyte aggregation, glomerular fibrin thrombi with ischemic collapse, and prominent expression of ACE2 staining on proximal tubular cells have been noted from autopsy findings on deceased COVID-19 patients [21], implying that this tough virus may also attack the kidney. The incidence of AKI seems to be positively correlated to disease severity, especially in patients with a respiratory failure complication. A large retrospective study indicated that AKI occurred in approximately 90% of COVID-19 patients who required ventilation [5]. In our hospital, only one patient (a 60-year-old woman with hypertension) presented with respiratory failure due to severe pneumonia and acute respiratory distress syndrome. She was excluded from our study because of her critical condition. Even though none of our patients met the criteria of AKI, we found a significant elevation of eGFR during hospitalization. A number of factors will lead to a decline in creatinine concentration, such as reduction of muscle mass, vegetarian diet, and hyperthyroidism-related glomerular hyper-ultrafiltration [22]. However, all patients were neither bedridden nor vegetarian. Only one patient had hyperthyroidism with medical control, but she did not present with other clinical manifestations of severe hyperthyroidism. According to the higher mortality risk associated with eGFR below 60 mL/min/1.732 m^2^, KDIGO defines eGFR lower than 60 mL/min/1.732 m^2^ as kidney impairment, regardless of age [23]. The decline of GFR with aging is thought to result from deterioration in renal structure, which usually begins after 30–40 years of age [24]. Concerning the relationship between GFR and age, elderly people may have less muscle mass and be misleading in the interpretation of GFR. Thus, samples in the MDRD equation are thought to be limited to patients under 70 years [25]. In our study, there were only two patients older than 70 years. The first was a 71-year-old man; his BMI was 25.51 kg/m^2^ and eGFR was 78.1 mL/min/1.732 m^2^ upon admission. His eGFR increased to 110 mL/min/1.732 m^2^ at discharge; this result was compatible with the results of most of other cases. The second patient was an 81-year-old man with a history of chronic kidney disease for 2 years; his BMI was 26.45 kg/m^2^ and eGFR was 61.8 mL/min/1.732 m^2^ upon admission. His eGFR decreased to 56.3 mL/min/1.732 m^2^ at discharge, which was similar to the level of his previous renal function.

So, what can explain the improvement of renal function? First and foremost, it does not make sense that patients have improved renal function following SARS-CoV-2 infection, which mostly promotes kidney injury. Furthermore, the timing of AKI presentation in most critical cases develops early in the course, occurring in approximately 37.3% of cases within 24 h of admission [5]. Thus, we assumed that mild renal function impairment would occur in early SARS-CoV-2 infection, even in patients with mild symptoms or those who are asymptomatic. Aitor Uribarri et al. demonstrated the association between creatinine clearance and the probability of all-cause death. The result showed that creatinine clearance had a negative correlation with predicted mortality, and this trend was noted when eGFR was below 150 mL/min/1.732 m^2^ [26]. Although the dynamic changes of renal functions in asymptomatic COVID 19 patients are still unknown, we believe slight difference in GFR still play an important role, regardless of whether the patient’s renal impairment meets the definition of AKI.

Dehydration is an important and common risk factor for renal damage, while mild dehydration is usually asymptomatic and under-detected. Unfortunately, our patients only measured their body weight once upon admission because all could move freely and were thought to be relatively healthy individuals. We excluded the probability of severe dehydration impacting renal function in the patients in our study. Although we did not perform accurate assessments of patients’ volume status, some patients might have had mild dehydration upon admission, which could present in several situations and was common in active viral infection. A large retrospective study had disclosed that sub-morbid dehydration may lead to glomerular hyperfiltration, presenting with elevated eGFR [27]. Thus, we presumed that changes of volume status had a lower degree of correlation with the downregulation of eGFR in our study.

We also found that the degree of kidney injury may be correlated with the degree of early infection. SARS-CoV-2 infection can activate innate and adaptive immune responses, and several immune-inflammatory parameters can imply the progression of illness. Prior studies showed that severe COVID-19 patients may have a higher neutrophil count and lower lymphocyte count; meanwhile, lymphopenia is also regarded as a predictor of prognosis in COVID-19 cases, and is more predominant in younger compared to older patients [28,29]. As mentioned above, SARS-CoV-2 mainly attacks the tissues expressing high levels of ACE2. Lymphocytes express the ACE2 receptor on their surface, so SARS-CoV-2 can directly infect those cells, leading to apoptosis [30]. Much evidence indicated NLR to be a better biomarker for predicting disease severity than single neutrophil or lymphocyte count [31,32]. Looking back on our study, we noted a statistically significant change in CRP, as well as in the number of neutrophils and lymphocytes during hospitalization, which implied the effect of infection control through medical treatment. Otherwise, a decline of renal function was found in patients with higher CRP, NLR, and neutrophil counts and lower lymphocyte counts upon admission. However, the same tendency was not noted in patients recovering from COVID-19. Thus, we thought that the extent of renal impairment was strongly related to the severity of virus infection in the early phase, regardless of whether or not those patients had symptoms.

Severe acute respiratory illness with fever and respiratory symptoms, such as cough and dyspnea, are the most widely known clinical manifestations. However, many patients still have non-respiratory symptoms, including gastrointestinal discomfort, diarrhea, and taste and olfactory disorders [33]. Some patients, especially in the dialysis population, only present with a flu-like illness, even without a fever, which may cause difficulty in early diagnosis, probably causing the rapid spread [34]. Our study demonstrated that the most common clinical presentation was cough, but only two out of three patients had fever. Approximately 40% of patients presented with atypical symptoms. In contrast to other international studies that discussed the impact of ethnicity on clinical outcomes, nearly all the patients admitted to our hospital developed mild symptoms or were even asymptomatic. Five patients did not feel any discomfort but underwent nasopharyngeal swab screening simply because they had close contacts with confirmed cases.

Viral infections cause kidney damage through direct invasion leading to cytopathic injury or through effects mediated by inflammation. Although the majority of virus infections may induce systemic immune responses, not all viruses may invade the kidney directly [35]. The etiology of AKI in COVID-19 cases has not been fully elucidated. Several pathophysiologies had been assumed to explain the relationship between SARS-CoV-2 and AKI. The cause of kidney involvement in COVID-19 is prone to be multifactorial, including maladaptive systemic inflammatory immune response, cytokine release syndrome, rhabdomyolysis, hypercoagulability, and the generation of microthrombi, leading to acute ischemia and AKI [36,37,38]. Otherwise, histopathologic examination from autopsied COVID-19 kidney tissue showed viral particles in podocytes and proximal renal tubules combined with varying degrees of acute tubular necrosis and lymphocytic infiltration. SARS-CoV-2 may induce kidney damage through direct infection of renal parenchyma mediated by ACE2, leading to protein leakage, acute tubular necrosis, collapsing glomerulopathy, and mitochondrial impairment [21]. Hence, SARS-CoV-2 is unique and different from other viral infections and definitely affects renal tissue directly. Recent experimental data and the evolving concept of organ crosstalk have become topics of lively discussion. Cardiomyopathy and alveolar damage are common complications in COVID-19 patients with severe sepsis, and ensuing hypoxia and hypoperfusion may cause further tubular damage [38]. Coronavirus entry into host target cells requires the fusion of the viral envelope with cellular membranes. Speaking of the mechanism for SARS-CoV-2 infection in the kidney, ACE2 and transmembrane protease serine 2 (TMPRSS2) play important roles [39]. SARS-CoV-2 utilizes ACE2 as a receptor for cellular entry and engages TMPRSS2 for spike protein priming. SARS-CoV-2 seems to not only obtain initial entry via ACE2, but also subsequently down-regulates ACE2 expression. The absence of ACE2 results in ensuing angiotensin II overproduction, which may trigger complement activation, Toll-like receptor 4 (TLR4) activation, hypercoagulability, and microangiopathy. Based on this scenario, we assume that the renin–angiotensin–aldosterone system (RAAS) inhibitor may affect ACE2 levels and activity [40,41,42]. According to medical records, many patients were young people, and only two patients had hypertension, both of whom took 5 mg amlodipine for blood pressure control. Thus, the issue of the additional effects of RAAS blocker on COVID-19 patients did not exist in our study.

Around one-third of patients were treated with temporary intravenous antibiotics due to single or multiple local ground-glass opacification of the lung on HRCT. Most of the other patients only took chloroquine with or without a macrolide for about 2 weeks. Although chloroquine is found to be a potent inhibitor of SARS-CoV-2 in vitro, current studies do not support its clinical effectiveness in COVID-19 patients [43,44]. Chloroquine has demonstrated adverse effects on kidney in rodent models, which may cause glomerular damage and tubular injury with drug accumulation [45]. As mentioned above, SARS-CoV-2 does attack podocytes as well as proximal renal tubules, and may result in tubular necrosis and further glomerulopathy via the ACE2-mediated pathway. Thus, synergism between SARS-CoV-2 infection and chloroquine combination can worsen renal toxicity.

Although patients included in this study only presented with mild illness or remained asymptomatic, they were not the same as patients with minor viral infections. Asymptomatic individuals may have a weaker immune response to SARS-CoV-2 infection but some have been found to have high viral loads. It might be the invisible part of the iceberg because prolonged shedding of SARS-CoV-2 RNA in the respiratory tract has been observed in convalescent patients and did not correlate with clinical infectivity [46,47,48]. In a previous systemic review, it was noted that the length of stay (LOS) may vary by country according to different discharge criteria. A longer LOS was found in China due to the requirement that patients have two negative PCR tests for discharge [49]. There is broad consensus that performing several repeated tests is recommended to overcome an individual test’s limited sensitivity if high pretest probability presents [50]. The Taiwanese CDC has a more restrictive discharge policy. Three consecutive negative virus RT-PCR results are indicated prior to discharge. Therefore, prolonged hospitalization is observed in Taiwan.

## 5. Limitations

Our study still had several limitations. First, it had a small sample size because it was conducted at a single medical center in northern Taiwan. Second, because this was a retrospective observational study and all COVID-19 patients were admitted to the negative pressure isolation ward division of infectious disease, most patients did not undergo dipstick urinalysis during hospitalization, not to mention urine chemistries. Nor were we able to analyze the prevalence of hematuria and proteinuria or utilize urinary excretion of electrolytes to evaluate volume status and clinical conditions. Third, these patients did not undergo frequent tests for renal function. Because most of our patients presented with mild illness, laboratory tests were only performed twice: once on admission day and again with samples collected within 72 h prior to discharge. Thus, the velocity of kidney recovery was difficult to understand. Finally, our patients only had their body weight measured once, on admission day. We could not determine whether a change in weight or muscle mass during their hospital stays might have been related to the decreased production of creatinine.

## 6. Conclusions

In conclusion, the kidney is also an emerging target for COVID-19. We demonstrated that dynamic changes of eGFR correlated with the elevation in the neutrophil count, NLR, and CRP. Renal involvement was strongly associated with higher mortality. Our findings inferred that renal impairment may also develop at an early stage of infection in asymptomatic patients or those with only mild symptoms. The biomarkers (CRP and NLR) may be linked with dynamic changes of renal function in COVID-19 patients with asymptomatic and mild symptoms.

## Figures and Tables

**Figure 1 jpm-11-00432-f001:**
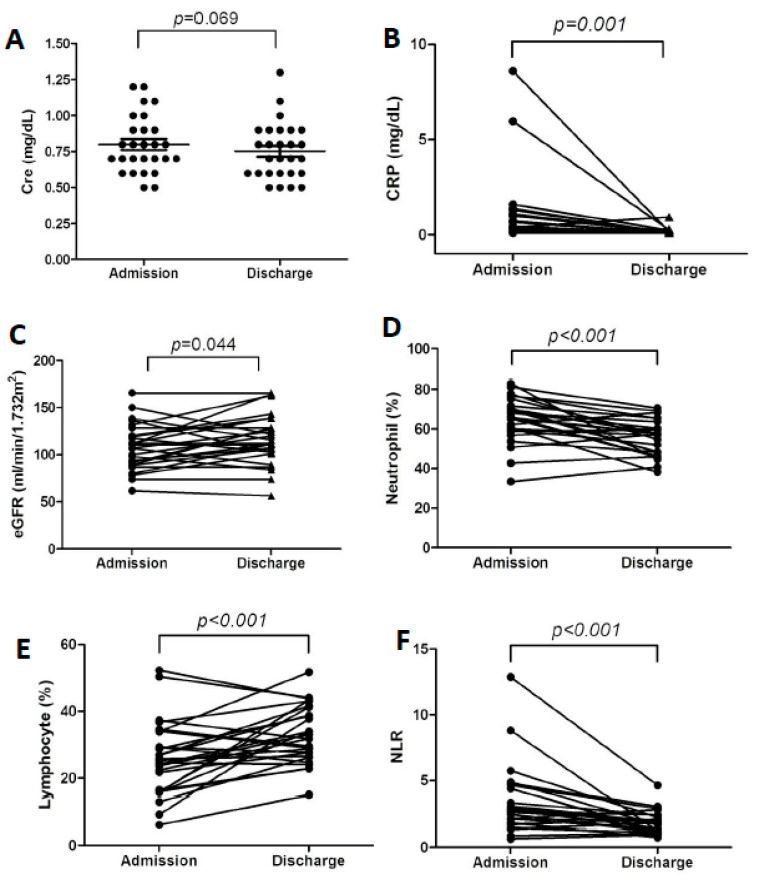
Comparison of data collected upon hospital admission and discharge. (**A**) Serum creatinine. (**B**) CRP level. (**C**) eGFR. (**D**) Neutrophil count. (**E**) Lymphocyte count. (**F**) Neutrophil-to-lymphocyte ratio.

**Figure 2 jpm-11-00432-f002:**
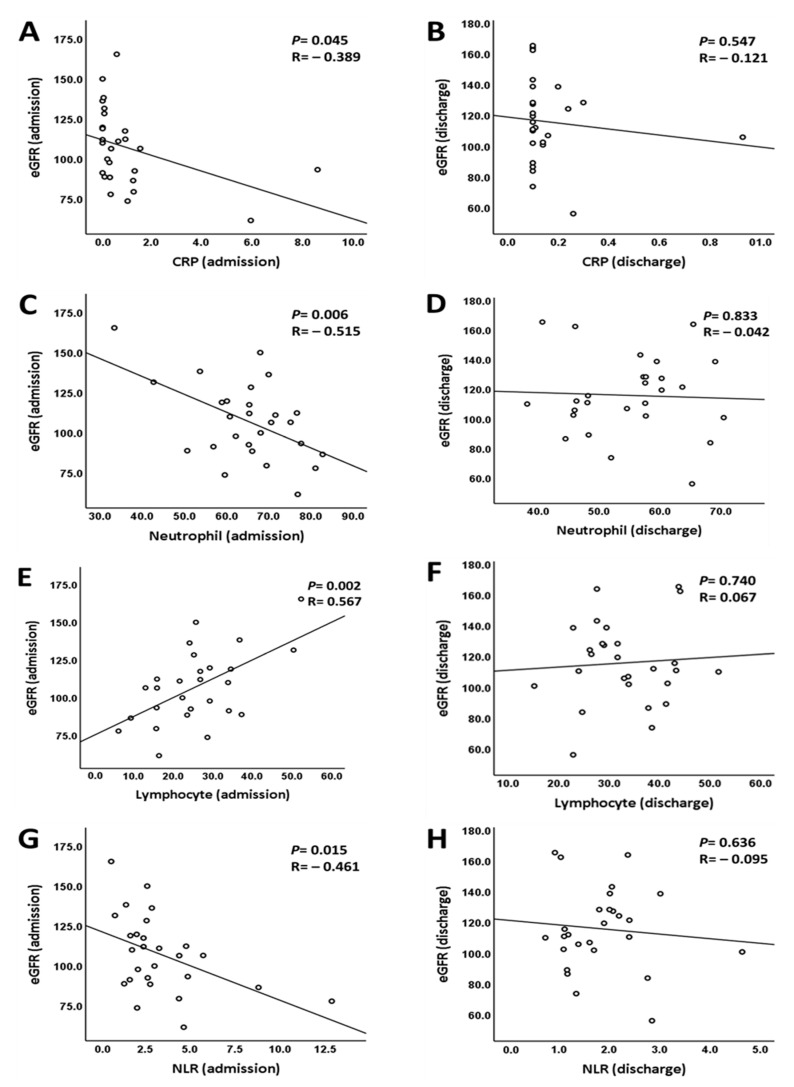
Correlation between renal function and the severity of infection. (**A**) Correlation between CRP and eGFR upon admission, the adjusted *p* value = 0.152 after adjustment for neutrophil and lymphocyte counts upon admission. (**B**) Correlation between CRP and eGFR at discharge. (**C**) Correlation between neutrophil count and eGFR upon admission, the adjusted *p* value = 0.240 after adjustment for CRP and lymphocyte count upon admission. (**D**) Correlation between neutrophil count and eGFR at discharge. (**E**) Correlation between lymphocyte count and eGFR upon admission, the adjusted *p* value = 0.071 after adjustment for lymphocyte count and CRP upon admission. (**F**) Correlation between lymphocyte and eGFR at discharge. (**G**) Correlation between NLR and eGFR upon admission, the adjusted *p* value = 0.037 after adjustment for CRP upon admission. (**H**) Correlation between NLR and eGFR at discharge.

**Table 1 jpm-11-00432-t001:** Clinical characteristics (*n* = 27).

Variables	N (%) or Median (IQR)
Age (years)	34 (23–54)
Male	15 (55.6%)
BMI (kg/m^2^)	21.8 (20.17–23.94)
SBP (mmHg)	130 (119–137)
DBP (mmHg)	80 (76–86)
Cormobidities	
Diabetes	2 (7.4%)
Hypertension	2 (7.4%)
Dyslipidemia	7 (25.9%)
Hepatic cirrhosis	1 (3.7%)
Coronary artery disease	1 (3.7%)
Hyperthyroidism	1 (3.7%)
Heart failure	0 (0%)
COPD	0 (0%)
Cerebrovascular accident	0 (0%)
Malignancy	1 (3.7%)
Symptoms on admission	
Cough	23 (85.2%)
Fever	17 (62.9%)
Rhinorrhea	12 (44.4%)
Abnormal sense of smell	11 (40.7%)
Diarrhea	10 (37.0%)
Shortness of breath	9 (33.3%)
Sore throat	9 (33.3%)
Abnormal sense of taste	8 (29.6%)
Chillness	5 (18.5%)
Myalgia	5 (18.5%)
Chest pain	5 (18.5%)
Headache	4 (14.8%)
Fatigue	4 (14.8%)
Nausea/Vomiting	3 (11.1%)
Asymptomatic	5 (18.5%)

IQR: interquartile range. BMI: body mass index. SBP: systolic blood pressure. DBP: diastolic blood pressure. COPD: chronic obstructive pulmonary disease.

**Table 2 jpm-11-00432-t002:** Disease severity and complications (*n* = 27).

Variables	N (%) or Median (IQR)
Hospital stay (day)	28 (21–35)
CURB-65	
0	23 (85.2%)
1	3 (11.1%)
2	1 (3.7%)
Chest images	
Ground-glass opacity	12 (44.4%)
No pneumonia	15 (55.5%)
Complications	
Hepatitis	4 (14.8%)
Multiple organ dysfunction	0 (0%)
Pulmonary fibrosis	0 (0%)
DIC	0 (0%)
Acute kidney injury	0 (0%)
Septic shock	0 (0%)
Myocarditis	0 (0%)
Acute coronary syndrome	0 (0%)
Cardiac arrest	0 (0%)

IQR: interquartile range. DIC: disseminated intravascular coagulation.

**Table 3 jpm-11-00432-t003:** The main medical treatment (*n* = 27).

Variables	N (%) or Median (IQR)
Chloroquine	20 (74.0%)
Macrolide antibiotics	15 (55.5%)
Floroquinolones	9 (33.3%)
Cephalosporin	8 (29.6%)
Piperacillin/Tazobactam	2 (7.4%)
Corticosteroids	1 (3.7%)
No treatment	5 (18.5%)

**Table 4 jpm-11-00432-t004:** Laboratory findings (*n* = 27).

Parameters	On Admission [Median (IQR)]	On Discharge [Median (IQR)]	*p*-Value ^a^
Hemoglobin (g/dL)	14.3 (13.2–15.1)	13.9 (12.1–15.0)	0.006
WBC (cells/µL)	4760 (4070–6700)	5680 (4980–6630)	0.027
Platelet (cells/µL)	230,000 (194,000–272,000)	233,000 (220,000–290,000)	0.516
Neutrophil (%)	65.8 (59.6–71.6)	57.3 (46.3–60.3)	<0.001
Monocyte (%)	6.7 (5.3–8.5)	6.5 (5.5–7.4)	0.243
Lymphocyte (%)	25.3 (16.0–33.9)	31.7 (26.5–41.3)	0.001
NLR (%)	2.64 (1.79–4.41)	1.81 (1.17–2.37)	<0.001
BUN (mg/dL)	12 (10–14)	11 (10–13)	0.455
Creatinine (mg/dL)	0.8 (0.7–0.9)	0.7 (0.6–0.9)	0.069
eGFR (mL/min/1.732 m^2^)	106.7 (89.0–119.9)	112.2 (102.1–128.5)	0.044
CRP (mg/dL)	0.39 (0.1–1.09)	0.1 (0.1–0.14)	0.001
Albumin (g/dL)	4.2 (3.98–4.43)	-	-
AST (U/L)	18 (14–21)	17 (14–23)	0.946
ALT (U/L)	15 (12–21)	20 (10–31)	0.095
Na (mmol/L)	138 (137–140)	139 (138–140)	0.497
K (mmol/L)	3.6 (3.5–4.0)	3.8 (3.7–4.1)	0.088

IQR: interquartile range. WBC: white blood cell. NLR: neutrophil-to-lymphocyte ratio. BUN: blood urea nitrogen. eGFR: estimated glomerular filtration rate. CRP: C-reactive protein. AST: aspartate aminotransferase. ALT: alanine aminotransferase. ^a^ Wilcoxon signed rank test.

**Table 5 jpm-11-00432-t005:** The role of NLR in COVID-19.

Author (Year)	Region	Study Period	Study Population	Biomarker	Positive Correlation
Prisca Mutinelli-Szymanski et al. (2021) [9]	France	19 March 2020, to 19 May 2020	62 dialysis patients	NLR (Day 7)	COVID-19 severity
Sara Jimeno et al. (2021) [10]	Spain	1 March 2020, to 31 March 2020	119 hospitalized patients	NLR	COVID-19 progression
Nicholas L Hartog et al. (2021) [11]	USA	20 March 2020, to 18 May 2020	66 hospitalized patients receiving tocilizumab	NLR	Unfavorable outcomes (intubation or mortality)
Mehr Muhammad Imran et al. (2021) [12]	Pakistan	1 May 2020, to 31 July 2020	63 hospitalized patients	NLR	Early warning signal for deteriorating severe COVID-19 infection
Gaoli Liu et al. (2020) [13]	China	28 January 2020, to 15 March 2020	134 hospitalized patients with type 2 diabetes mellitus	NLR	1. COVID-19 severity 2. Timing of nucleic acid results turned negative 3. Duration of hospital stay
Jianhong Fu et al. (2020) [14]	China	20 January 2020, to 20 February 2020	75 hospitalized patients	NLR	Discriminate severe COVID-19 cases from mild or moderate ones
Our study	Taiwan	4 February 2020, to 26 May 2020	27 hospitalized patients	NLR (Day 1)	Decline of renal function

NLR: neutrophil-to-lymphocyte ratio. COVID-19: coronavirus disease 2019.

**Table 6 jpm-11-00432-t006:** The role of CRP in COVID-19.

Author (Year)	Region	Study Period	Study Population	Biomarker	Positive Correlation
Dominic Stringer et al. (2021) [15]	United Kingdom	27 February 2020, to 10 June 2020	1835 hospitalized patients	CRP ≥40 mg/L	Mortality
Nathaniel R. Smilowitz et al. (2021) [16]	USA	1 March 2020, to 8 April 2020	2782 hospitalized patients	CRP	1. Venous thromboembolism 2. Acute kidney injury 3. Critical illness 4. Mortality
Milad Sharifpour et al. (2020) [17]	USA	6 March 2020, to 5 May 2020	268 ICU patients	CRP	1. Disease severity 2. Mortality
Xiaomin Luo et al. (2020) [18]	China	30 January 2020, to 20 February 2020	298 hospitalized patients	CRP	Mortality
Chaochao Tan et al. (2020) [19]	China	18 January 2020, to 10 February 2020	27 hospitalized patients	CRP	1. Disease development 2. CT severity score
L Wang (2020) [20]	China	23 January 2020, to 29 February 2020	27 hospitalized patients	CRP	1. Diameter of lung lesion 2. Disease severity
Our study	Taiwan	4 February 2020, to 26 May 2020	27 hospitalized patients	CRP (Day 1)	Decline of renal function

CRP: C-reactive protein. ICU: intensive care unit. CT: computed tomography.

## Data Availability

Restrictions apply to the availability of these data. Data were obtained from electronic medical records in the Tri-Service General Hospital database.

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
