# Peer review of "Biomarkers Linked with Dynamic Changes of Renal Function in Asymptomatic and Mildly Symptomatic COVID-19 Patients"

_jpm, 2021, doi:10.3390/jpm11050432_

Round 1

Reviewer 1 Report

In this MS, Chang et al have reported the impact of renal dynamics in asymptomatic COVID-19 patients. While the pandemic is still ongoing and growing, any sort of new information will definitely be useful to the clinicians and in this retrospective, single-centre study authors have reported the renal function impairment from a smallcohort of patients, which subtly dampens the enthusiasm. Authors also need to pay more attention to the statistics. Authors are highly recommended and required to include another cohort of patients of equal number or more to make an authoritative conclusion, which is a major comment for this MS. Here are some more comments –

  1. Authors report a decreased eGFR and their correlation with systemic inflammation markers (NLR, lymphocyte, CRP etc.) on admission and discharge. They showed a high correlation of eGFR with these markers on admission but no correlation on discharge is explained.
  2. Conclusions – It needs a rewrite. Conclusion appears to be a descriptive discussion.
  3. Based on the findings, authors should not be concluding that ’higher CRP and NLR can predict the worsening of renal function’ because they only did correlation analysis.
  4. Because, eGFR correlates with age, what’s the relation between eGFR and age in this study? Need a detailed analysis and discussion. How many patients did have abnormal eGFR in this study (calculated by their age, BMI and body surface area etc.)?
  5. Because no patients represented AKI in this study, how could your marker show a correlation with (or predict) renal impairment?
  6. Before statistical analysis, data should be tested for distribution and variance. Non-normal distributed variables should not be listed as Mean±SD. For comparisons between - on admission and discharge data, authors should compare with paired t-tests or Wilcoxon signed-ranked tests.
  7. For continuous variable, authors should evaluate whether the data is normally distributed and then decide to use average or median number.
  8. Authors have performed whole lot of correlations in the entire study. Were those adjusted for multiplicity? If not, authors are required to do that and report the adjusted p values.
  9. Table 4-5. Here in, authors have used different method for data analysis but some of them are case report. Hence, it can’t be concluded as ‘positive correlation’. Authors need to be fix this.
  10. Authors are required to provide a table about the treatment regimen of the patients and also state, whether the treatment was nephrotoxic?
  11. Most importantly, authors need to increase the number of patient cohort.

Author Response

For reviewer #1:

We are deeply honored by the time and effort you spent in reviewing this manuscript. We have revised the manuscript thoroughly according to your suggestions. The responses to your comments are below.

  1. Authors report a decreased eGFR and their correlation with systemic inflammation markers (NLR, lymphocyte, CRP etc.) on admission and discharge. They showed a high correlation of eGFR with these markers on admission but no correlation on discharge is explained.

Ans.: Yes, we have founded this point. We found that the degree of kidney injury may be correlated with the degree of early infection. SARS-CoV-2 infection can activate innate and adaptive immune responses, and several immune-inflammatory parameters can imply the progression of illness. Prior studies showed that severe COVID-19 patients may have a higher neutrophil count and lower lymphocyte count; meanwhile, lymphopenia is also regarded as a predictor of prognosis in COVID-19 cases. As mentioned above, SARS-CoV-2 mainly attacks the tissues expressing high levels of ACE2. Lymphocytes express the ACE2 receptor on their surface, so SARS-CoV-2 can directly infect those cells, leading to apoptosis. Many evidences indicated NLR to be a better biomarker for predicting disease severity than single neutrophil or lymphocyte count. Looking back on our study, a statistical significance in the change of CRP, as well as the number of neutrophils and lymphocytes during hospitalisation, was noted, which implied the effect of infection control through medical treatment. Otherwise, a decline of renal function was found in patients with higher CRP, NLR and neutrophil count and lower lymphocyte count upon admission. But the same tendency was not noted when recovering from COVID-19. Thus, we thought that the extent of renal impairment is strongly related to the severity of virus infection in the early phase, regardless of whether or not those patients have symptoms. 

Please see Discussion, lines 310-327.

  1. Conclusions – It needs a rewrite. Conclusion appears to be a descriptive discussion.

Ans.: We have rewritten the conclusion to make it more concise and conclusive in our revised manuscript.  Please see Conclusion, lines 412-418.

  1. Based on the findings, authors should not be concluding that ’higher CRP and NLR can predict the worsening of renal function’ because they only did correlation analysis.

Ans.: Yes, we totally agreed with your comments. Our findings definitely cannot predict the worsening of renal function, but only correlated. Hence, we revised it according to reviewer’s suggestion. Please see Results 3.4, lines 170-175

  1. Because, eGFR correlates with age, what’s the relation between eGFR and age in this study? Need a detailed analysis and discussion. How many patients did have abnormal eGFR in this study (calculated by their age, BMI and body surface area etc.)?

Ans.: According to the higher mortality risk associated with eGFR below 60 ml/min/1.732 m2, KDIGO define eGFR lower than 60 ml/min/1.732 m2 as kidney impairment, regardless of age. The decline of GFR with aging is thought to be resulting from deterioration in renal structure, which usually begin after 30–40 years of age. Concerning the relationship between GFR and age, elder people may have less muscle mass and be misleading in the interpretation of GFR. Thus, sample in the MDRD equation are thought to be limited to patients under 70 years. In our study, there were only two cases older than 70 years. The first one was a 71-year-old man, his BMI was 25.51 kg/m2 and eGFR was 78.1 ml/min/1.732 m2 upon admission. His eGFR increased to 110 ml/min/1.732 m2 at discharge, this result was compatible with the results of most of other cases. The second case was a 81-year-old man with history of chronic kidney disease for 2 years, his BMI was 26.45 kg/m2 and eGFR was 61.8 ml/min/1.732 m2 upon admission. His eGFR decreased to 56.3 ml/min/1.732 m2 at discharge, which was similar to his previous renal function. Please see Discussion, lines 272-285.

  1. Because no patients represented AKI in this study, how could your marker show a correlation with (or predict) renal impairment?

Ans. The dynamic changes of renal functions in asymptomatic COVID 19 patients is still unknown. We believe that slight difference in GFR in asymptomatic COVID-19 patients is meaningful, regardless of whether the patient’s renal impairment meet the definition of AKI. Our study showed that renal impairment may also develop at an early stage of COVID 19 infection in patients with mild symptoms. The renal involvement was strongly associated with higher mortality. COVID-19-associated AKI (COVID-19 AKI) is associated with high mortality and serves as an independent risk factor for all-cause in-hospital death in patients with COVID-19. (Ref. 1, Nat Rev Nephrol. 2020 Dec;16(12):747-764.) According to the results from international HOPE COVID-19 (Health Outcome Predictive Evaluation for COVID 19) Registry, renal function at admission behaved as an independent prognostic factor. The impact of creatinine clearance on mortality is displayed graphically as follows Fig. 2. As creatinine clearance decreased, the probability of death increased. Importantly, when we excluded patients with known chronic renal failure from the analysis the results remained consistent. (Ref. 2, J Nephrol. 2020 Aug;33(4):737-745.). We demonstrated that the dynamic changes of eGFR correlated with the elevation in the neutrophil count, NLR and CRP. Our findings inferred that renal impairment may also develop at an early stage of infection in asymptomatic patients or those with only mild symptoms, even in young people. Hence, we thought that our marker show a correlation with renal impairment.

Please see Discussion, lines 292-298 .

 (Ref. 2, J Nephrol. 2020 Aug;33(4):737-745.)

Ref. 1 Mitra K Nadim, Lui G Forni, Ravindra L Mehta et al. COVID-19-associated acute kidney injury: consensus report of the 25th Acute Disease Quality Initiative (ADQI) Workgroup. Nat Rev Nephrol. 2020 Dec;16(12):747-764.

Ref. 2 Aitor Uribarri, Iván J Núñez-Gil, Alvaro Aparisi et al. Impact of renal function on admission in COVID-19 patients: an analysis of the international HOPE COVID-19 (Health Outcome Predictive Evaluation for COVID 19) Registry. J Nephrol. 2020 Aug;33(4):737-745.

  1. Before statistical analysis, data should be tested for distribution and variance. Non-normal distributed variables should not be listed as Mean±SD. For comparisons between on admission and discharge data, authors should compare with paired t-tests or Wilcoxon signed-ranked tests.

Ans.: Thank you for your comments. We have presented as median (interquartile range, IQR) for continuous measurements because of non-normal distribution. We also have compared with Wilcoxon signed-ranked tests for comparisons between on admission and discharge data.

Please see Materials and Methods 2.3, line 107-112 and Table 1,2,4.

  1. For continuous variable, authors should evaluate whether the data is normally distributed and then decide to use average or median number.

Ans.: Thank you for your comments. We have presented as median (interquartile range, IQR) for continuous measurements because of non-normal distribution.

Please see Materials and Methods 2.3, line 107-112 and Table 1,2,4

  1. Authors have performed whole lot of correlations in the entire study. Were those adjusted for multiplicity? If not, authors are required to do that and report the adjusted p values.

Ans.: Thank you for your comments. We have completed multiplicity analysis and the adjusted p values have presented.  Please see Figure 2 legend.

  1. Table 4-5. Here in, authors have used different method for data analysis but some of them are case report. Hence, it can’t be concluded as ‘positive correlation’. Authors need to be fix this.

Ans. Thanks for your reminders. We have removed the case report and added correct reference in our revised manuscript. Please see Table 5, Ref 12.

  1. Authors are required to provide a table about the treatment regimen of the patients and also state, whether the treatment was nephrotoxic?

Ans. Thanks for your valuable suggestion. We have added a table about the treatment regimen of the patients. Please see Table 3, Results 3.2, lines 140-143 and Discussion, line 372-382.

  1. Most importantly, authors need to increase the number of patient cohort.

Ans.: Thank you for pointing out the critical weakness, small sample size, in our study. We cannot collect as much number cases as those in other countries under Taiwan’s success of the epidemic control. We totally agreed that larger sample size may strength the power in our study. Although the sample size is small, we think our findings are novel and providing significances and values for clinical application. We have added it as limitation. Please see Limitation, lines 398-399.

Reviewer 2 Report

The title does not accurately represent the patients included in this study. Patients with mild illness were included rather than only asymptomatic patients.

85% of patients had cough suggesting these patients were symptomatic.

What were the criteria for discharge?

The mean LOS was 28 days. The authors should explain why the mean was so high if most of these patients were asymptomatic or had only mild illness.

The design can be improved by including data at the midpoint of the hospitalization. Given the prolonged hospitalization, it is possible that the patients may have suffered renal injury during the hospitalization which resolved prior to discharge.

It is possible that the GFR change between admission and discharge may be due to better hydration and monitoring during hospitalization.

The initial cr and GFR on admission may reflect the dehydration status of the patients rather than the severity of the disease.

Author Response

For reviewer #2:

Many thanks for the time and effort you spent in reviewing this manuscript. We have revised the manuscript thoroughly according to your suggestions. The responses to your comments are below.

  1. The title does not accurately represent the patients included in this study. Patients with mild illness were included rather than only asymptomatic patients.

85% of patients had cough suggesting these patients were symptomatic.

Ans. We have adjusted the title to Biomarkers linked with dynamic changes of renal function in asymptomatic and mildly symptomatic COVID-19 patients.

  1. What were the criteria for discharge?

Ans. According to the Taiwan CDC guideline, the discharge criteria include clinical improvement with radiographic resolution and three consecutive negative virus RT-PCR results. Please see Materials and Methods 2.1, lines 88-90.

  1. The mean LOS was 28 days. The authors should explain why the mean was so high if most of these patients were asymptomatic or had only mild illness.

Ans. Although patients included in this study only presented with mild illness or remained asymptomatic, they were not the same as minor viral infection. Asymptomatic individuals may have a weaker immune response to SARS-CoV-2 infection but high viral loads are found among them. It might be the invisible part of the iceberg because prolonged shedding of SARS-CoV-2 RNA in respiratory tract had been observed in convalescent patients, and did not correlate with clinical infectivity. In previous systemic review, the length of stay (LOS) may be various according to different discharge criteria in the countries. The longer LOS is found in China due to two negative PCR test is required for discharge. There is broad consensus that performing several repeated tests is suggested for overcoming an individual test’s limited sensitivity if high pretest probability presents. The Taiwanese CDC is more restricted to discharge policy. Three consecutive negative virus RT-PCR results are indicated prior to discharge. Therefore, prolonged hospitalization is observed in Taiwan. Please see Discussion, lines 383-395.

  1. The design can be improved by including data at the midpoint of the hospitalization. Given the prolonged hospitalization, it is possible that the patients may have suffered renal injury during the hospitalization which resolved prior to discharge.

Ans. We totally agreed with your points that frequent measurement may improve our study. Unfortunately, our patients only measured twice, once on the admission day and another on discharge because most of our patients were asymptomatic or only presented with mild symptom on admission day. Hence, there is no data available. We added these statements as the limitation. Please see Limitation, lines 405-408.

  1. It is possible that the GFR change between admission and discharge may be due to better hydration and monitoring during hospitalization.

   The initial cr and GFR on admission may reflect the dehydration status of the   

   patients rather than the severity of the disease.

Ans. Dehydration is an important and common risk factor for renal damage, while mild dehydration was usually asymptomatic and under-detected. Unfortunately, our patients only measured their body weight once upon admission because all of these cases could move freely and were thought to be relatively healthy individuals. We excluded the probability of severe dehydration impacting on renal function occurred in our cases. Although we did not perform accurate assessment of patients’ volume status and some cases might have mild dehydration upon admission, which could present in several situations and was common in active viral infection. A large retrospective study had disclosed that sub-morbid dehydration may lead to glomerular hyperfiltration, presenting with elevated eGFR. Thus, we supposed that changes of volume status had a lower degree of corre-lation with the downregulation of eGFR in our study. Please see Discussion, lines 299-309.

Round 2

Reviewer 1 Report

No more comments

Author Response

For reviewer #1:

We are deeply honored by the time and effort you spent in reviewing this manuscript. Thanks for your positive evaluation and comments.

Reviewer 2 Report

Authors would benefit from addition of mid-hospitalizations parameters and addressing prior recommendations

Author Response

For reviewer #2:

Many thanks for the time and effort you spent in reviewing this manuscript. We have revised the manuscript thoroughly according to your suggestions. The responses to your comments are below.

  1. Authors would benefit from addition of mid-hospitalizations parameters and addressing prior recommendations.

Ans.: We completely agreed with your suggestion that addition of mid-hospitalizations parameters may be beneficial. As the reviewer pointed out that it is possible that the patients may have suffered renal injury during the hospitalization which resolved prior to discharge or it is possible that the GFR change between admission and discharge may be due to better hydration and monitoring during hospitalization. However, most of our patients were asymptomatic or only presented with mild symptoms on admission day. The patients only measured laboratory tests twice, once on the admission day and another on discharge. Volume status of the patient has a major influence of renal functions. Body weight measurement also can provide valuable information. Unfortunately, our patients only measured their body weight once upon admission because all of these cases could move freely and were thought to be relatively healthy individuals. We excluded the probability of severe dehydration impacting on renal function occurred in our cases. Although we did not perform an accurate assessment of patients’ volume status and some cases might have mild dehydration upon admission. Systemic hemodynamics, such as blood pressure or heart rates may also reflect volume status. There were no significant changes of these systemic hemodynamic parameters throughout our study. We apologize that we cannot provide anymore mid-hospitalization parameters because this is a retrospective study. There were only four patients had mid-hospitalizations parameters assessing renal function and no dramatic changes were found in these cases. However, we totally agreed with your points and suggestion that frequent measurement of parameters such as lab or body weight are important for clinical practices to keep fluid status balance and avoid dehydration. We added these statements as the limitation and in the discussion section. We will include your excellent suggestion in our future study design.

Please see Discussion, line 298-308, Limitation, lines 404-409.

This manuscript is a resubmission of an earlier submission. The following is a list of the peer review reports and author responses from that submission.

Round 1

Reviewer 1 Report

Remarks to the Author

The authors examined whether COVID-19 affected the renal function in patients with mild symptoms. They found in this retrospective, single-center study that they can observe the decline in eGFR on admission day. Present findings are interesting and important; however, this study still has some concerns to be considered.

Major concerns:

  1. Figure1C, eGFR of some patients is downregulated on discharge day. What about focus on the characters of these patients?
  2. Table 3, Hemoglobin data is significantly lower at discharge day. This change suggests that they had dehydration on admission day. As you wrote in limitation, patients only measured their body weight once on the admission day. How do you think if patients measured their body weight more than once and keep avoiding to dehydration, the decline in eGFR in some patient on discharge day could avoid?
  3. The authors wrote “we thought that the extent of renal impairment is strongly related to the severity of virus infection in the early phase, regardless of whether or not those patients have symptoms.". Do you really think this virus directly affect renal tissue and cause this renal impairment?

minor concerns.

  1. I personally am interested in observing the long time eGFR change of patients whose eGFR was downregulated on discharge day.

Author Response

  1. Figure1C, eGFR of some patients is downregulated on discharge day. What about focus on the characters of these patients?

Ans.: Seven patients had downregulation of eGFR on discharge day. Among these cases, 4 patients had CURB65 score of 0; 2 patients had CURB65 score of 1; and 1, CURB65 score of 2. One patient (M/81) has diabetic nephropathy and chronic kidney disease, stage 3 primarily. Progressive reduction of renal function during this infection episode was observed in this case. The other six patients were found to have slight elevation of creatinine level (0.6 to 0.7 mg/dL in 4 cases and 0.7 to 0.8 mg/dL in 2 cases). We thought these patients may have weight gain owing to a lack of exercise during the hospitalization. However, we can’t prove it because all patients only measured their body weight once on the admission day. We added these statements in the results section. Please see Results 3.4, page 6, lines 174-179.

  1. Table 3, Hemoglobin data is significantly lower at discharge day. This change suggests that they had dehydration on admission day. As you wrote in limitation, patients only measured their body weight once on the admission day. How do you think if patients measured their body weight more than once and keep avoiding to dehydration, the decline in eGFR in some patient on discharge day could avoid?

Ans.: Definitely, dehydration is an important and common risk factor for renal damage. Mild dehydration was usually asymptomatic. We totally agreed with your points that frequently measurement of body weight is important for clinical practices to keep fluid status balance and avoid dehydration. We added these statements in the discussion section. Please see Discussion, page 8, lines 271-281.

  1. The authors wrote “we thought that the extent of renal impairment is strongly related to the severity of virus infection in the early phase, regardless of whether or not those patients have symptoms.". Do you really think this virus directly affect renal tissue and cause this renal impairment?

Ans.: The SARS‐CoV‐2‐induced kidney damage is expected to be multifactorial. Histopathologic examination from autopsied COVID-19 kidney tissue showed viral particles in podocytes and proximal renal tubules combined with varying degrees of acute tubular necrosis and lymphocytic infiltration. The SARS‐CoV‐2 may induce kidney damage through directly infect renal parenchyma mediated by ACE2 leading to protein leakage, acute tubular necrosis, collapsing glomerulopathy and mitochondrial impairment. The SARS‐CoV‐2 affects kidney also through dysregulation of the immune responses and the generation of microthrombi which leading to acute ischaemia and AKI. Virus directly affect renal tissue is definitely happened. We added these statements in the discussion section. Please see Discussion, page 9, lines 316-324.

Minor concerns.

  1. I personally am interested in observing the long time eGFR change of patients whose eGFR was downregulated on discharge day.

Ans.: Thanks for your comments and providing an interesting issue. The study to follow up the long time eGFR change of patients whose eGFR was downregulated on discharge day is ongoing.

Reviewer 2 Report

Comments:

This is an interesting study showing that renal impairment may also develop at an early stage of COVID 19 infection in patients with mild symptoms.

The authors made the observation that the eGFR declined with the elevation in the neutrophil count, neutrophil-to-lymphocyte ratio 169 (NLR) and CRP upon admission and an increase in eGFR and lymphocyte was noticed at discharge

From their findings, the authors inferred that renal impairment may also develop at an early stage of infection in asymptomatic patients or those with only mild symptoms, even in young people. Besides, the severity of infection may impact the degree of kidney injury. Higher CRP and NLR can predict the worsening of the renal function.

The article is neatly written, however I have a number of comments which the authors may wish to address though most are the limitations of the study

  1. Firstly volume status of the patient has a major influence of renal functions , however there is no data available

  1. It would be interesting to know /mention where there any systemic changes such as blood pressure, HR etc… that depend on renal function or in other word effect of the renal impairment on systemic parameters in this scenario would make the reading interesting.

  1. The title suggests “Biomarkers linked with dynamic changes of renal function in asymptomatic COVID-19 patients” but as such there are no real ‘biomarkers’ per say looked at, may be the title need improvisations – “Dynamic changes of renal functions inn asymptomatic COVID 19 patients” seems more better.

  1. Though the proposed hypothesis needs further evidence such as larger sample size, urine chemistry etc to clearly associate renal impairment to COVID 19 in asymptomatic patients, the available data has been neatly presented.

Author Response

For reviewer #2:

Many thanks for the time and effort you spent in reviewing this manuscript. We have revised the manuscript thoroughly according to your suggestions. The responses to your comments are below.

This is an interesting study showing that renal impairment may also develop at an early stage of COVID 19 infection in patients with mild symptoms.

The authors made the observation that the eGFR declined with the elevation in the neutrophil count, neutrophil-to-lymphocyte ratio 169 (NLR) and CRP upon admission and an increase in eGFR and lymphocyte was noticed at discharge

From their findings, the authors inferred that renal impairment may also develop at an early stage of infection in asymptomatic patients or those with only mild symptoms, even in young people. Besides, the severity of infection may impact the degree of kidney injury. Higher CRP and NLR can predict the worsening of the renal function.

The article is neatly written, however I have a number of comments which the authors may wish to address though most are the limitations of the study

 Ans.: Thank you for the positive evaluation and the excellent suggestions.

  1. Firstly volume status of the patient has a major influence of renal functions, however there is no data available

Ans.: We totally agreed with your points that volume status of the patient has a major influence of renal functions. Unfortunately, our patients only measured their body weight once on the admission day because most of our patients were asymptomatic or only presented with mild symptom on admission day. Hence, there is no data available. On the other hand, we also excluded the probability of severe dehydration impacting on renal function occurred in our cases. Usually, mild dehydration was asymptomatic and under-detected. The reviewer 1 also pointed the same point and suggested that frequently measurement of body weight is important for clinical practices to keep fluid status balance and avoid dehydration. We added these statements as the limitation and in the discussion section. Please see Discussion , page 8, lines 271-281 and Limitation, page 10, lines 350-352.

  1. It would be interesting to know /mention where there any systemic changes such as blood pressure, HR etc… that depend on renal function or in other word effect of the renal impairment on systemic parameters in this scenario would make the reading interesting.

Ans.: Thanks for your comments. It is an interesting question. Systemic hemodynamics such as blood pressure or heart rates may influence the renal function. However, on the other hand, renal impairment also affects these systemic parameters. In our study, most of our patients were asymptomatic or only presented with mild symptom on admission day, hence, there were lower probability of severe dehydration occurred in our cases. Although we cannot exclude the possibility of mild dehydration, there were no significant changes of these systemic parameters throughout our study under present renal functions.  Please see Results 3.1, page 3, lines 118.

  1. The title suggests “Biomarkers linked with dynamic changes of renal function in asymptomatic COVID-19 patients” but as such there are no real ‘biomarkers’ per say looked at, may be the title need improvisations – “Dynamic changes of renal functions in asymptomatic COVID 19 patients” seems more better.

Ans.: We have revised the Title as suggestion.

  1. Though the proposed hypothesis needs further evidence such as larger sample size, urine chemistry etc to clearly associate renal impairment to COVID 19 in asymptomatic patients, the available data has been neatly presented.

Ans.: Thank you for the positive evaluation and relevant comments. We agreed the proposed hypothesis needs further evidence such as larger sample size, urine chemistry etc to clearly associate renal impairment to COVID 19 in asymptomatic patients. We have added it as limitation. Please see Limitation, page 10, lines 342-348.

Reviewer 3 Report

The authors have presented data collected on a very small number of patients (27) and suggested  the change in GFR (despite no change in serum creatinine) is of some clinical significance. The proceeding discussion is based on this assumption and relates inflammatory markers to this change in GFR.

I feel that the change in GFR whilst perhaps numerically significant is of no clinical significance based on the data presented, and can  be explained by any viral illness symptoms. The authors have not related the GFR change in  to COVID in particular. I cannot see anything that supports the conclusions that this minor change in GFR is related to COVID and feel the data is of minor interest and use in the realm of COVID.

Author Response

For reviewer #3:

We are deeply honored by the time and effort you spent in reviewing this manuscript. We have revised the manuscript thoroughly according to your suggestions. The responses to your comments are below.

The authors have presented data collected on a very small number of patients (27) and suggested the change in GFR (despite no change in serum creatinine) is of some clinical significance. The proceeding discussion is based on this assumption and relates inflammatory markers to this change in GFR.

I feel that the change in GFR whilst perhaps numerically significant is of no clinical significance based on the data presented, and can be explained by any viral illness symptoms. The authors have not related the GFR change in to COVID in particular. I cannot see anything that supports the conclusions that this minor change in GFR is related to COVID and feel the data is of minor interest and use in the realm of COVID.

Ans.: The novel coronavirus (SARS‐CoV‐2) has turned into a life‐threatening pandemic disease (Covid‐19). SARS-CoV-2 binds to the angiotensin-converting enzyme 2 (ACE2) as a receptor for cellular entry, which abound in the type II alveolar cells of the lung. The kidney also highly expresses ACE2. Histopathologic examination from autopsied COVID-19 kidney tissue showed viral particles in podocytes and proximal renal tubules combined with varying degrees of acute tubular necrosis and lymphocytic infiltration. The COVID-19 induced kidney damages are involved through direct or indirect mechanisms. The virus may induce damage through directly infect renal parenchyma mediated by ACE2 leading to protein leakage, acute tubular necrosis, collapsing glomerulopathy and mitochondrial impairment. The SARS‐CoV‐2 affects kidney also through dysregulation of the immune responses and the generation of microthrombi which leading to acute ischaemia and AKI. Hence, the SARS‐CoV‐2 is unique and different from other viral infections. The dynamic changes of renal functions in asymptomatic COVID 19 patients is still unknown. Our study showed that renal impairment may also develop at an early stage of COVID 19 infection in patients with mild symptoms. We demonstrated that the dynamic changes of eGFR correlated with the elevation in the neutrophil count, neutrophil-to-lymphocyte ratio (NLR) and CRP. The renal involvement was strongly associated with higher mortality. Our findings inferred that renal impairment may also develop at an early stage of infection in asymptomatic patients or those with only mild symptoms, even in young people. Besides, we provided biomarkers (CRP and NLR) to predict the worsening of the renal function. Although the sample size is small, we think our findings are novel and providing significances and values for clinical application. Please see Discussion, page 9, lines 316-324 and Conclusion, page 10, lines 355-362.

Round 2

Reviewer 1 Report

Comments to the Author

Most of my comments are appropriately addressed.

Reviewer 3 Report

I am sorry but I am not convinced of the academic value of the publication. The slight difference in GFR demonstrated between admission and discharge n the small cohort may be of borderline statistical significance but is in my opinion of no clinical significance. The conclusion by the authors related to the effect of COVID on renal function are not justified by the data presented. I remain of the opinion that any minor viral is likely to have similar minor effect on GFR if any, and the observations are not particular to COVD based on the data presented. The postulated effects of COVID on the kidney suggested by the authors are not supported in any way by the data presented. I do not support the publication of this paper.